# ON THE EFFECTIVENESS OF DEEP ENSEMBLES FOR SMALL DATA TASKS

## ABSTRACT

Deep neural networks represent the gold standard for image classification. However, they usually need large amounts of data to reach superior performance. In this work, we focus on image classification problems with a few labeled examples per class and improve sample efficiency in the low data regime by using an ensemble of relatively small deep networks. For the first time, our work broadly studies the existing concept of neural ensembling in small data domains, through an extensive validation using popular datasets and architectures. We show that deep ensembling is a simple yet effective technique that outperforms current state-of-the-art approaches for learning from small datasets. We compare different ensemble configurations to their deeper and wider competitors given a total fixed computational budget and provide empirical evidence of their advantage. Furthermore, we investigate the effectiveness of different losses and show that their choice should be made considering different factors.

## 1 INTRODUCTION

The computer vision field has been revolutionized by the advent of deep learning (DL) [Y. LeCun & Hinton, 2015]. The convolutional neural network (CNN) is the most popular DL model for visual learning tasks thanks to its ability to automatically learn general features via gradient-based optimization algorithms. However, the cost to reach high recognition performances involves the collection and labeling of large quantities of images. This requirement can not always be fulfilled since it may happen that collecting images is extremely expensive or not possible at all. For instance, in the medical field, high-quality annotations by radiology experts are often costly and not manageable at large scales [Litjens et al., 2017]. Different approaches have been proposed by the research community to mitigate the necessity of training data, tackling the problem from different perspectives.

Transfer learning aims at learning representations from one domain and *transfer* the learned knowledge (e.g. pre-trained network) to a target domain [Bengio, 2012], [Tan et al., 2018]. Similarly, few-shot learning uses a *base set* of labelled pairs to generalize from a small *support set* of target classes [Vanschoren, 2018]. Both approaches suffer from the need of collecting a pool of annotated images and the *source* and *target* domains must be somewhat related. Self-supervised learning is another approach that is trying to reduce the demand for annotations. Usually, a large set of images is used to teach how to solve a *pretext* task to a CNN [Jing & Tian, 2020] in sight of teaching a later *downstream* task. In this manner, costly human annotations are not needed but the challenge of collecting many images remains.
The previously cited research directions, in a way or another, still rely on many samples/annotations. Our grand goal is to develop learning algorithms that are as sample-efficient as the human visual system. In other words, we aim to solve a classification problem with only a limited amount of labeled examples. Due to the great difficulty, this problem is still largely unsolved and hardly experimented.

In this work, we propose the use of neural ensembles composed of smaller networks to tackle the problem of learning from a small sample and show the superiority of such methodology. Similarly to what has been done in recent works [Arora et al., 2020], [Barz & Denzler, 2020], we benchmark the approaches by varying the number of data points in the training sample while keeping it low with respect to the current standards of computer vision datasets. It has been shown that large CNNs can handle overfitting and generalize well even if they are severely over-parametrized [Kawaguchi et al.,

2017], [Neyshabur et al., 2018]. A recent study have also empirically shown that such behaviour might also be valid in the case of tiny datasets, making large nets a viabile choice even when the training sample is limited [Bornschein et al., 2020]. A well-known technique to reduce model variance is to average predictions from a set of weak learners (e.g. random forests [Breiman, 2001]). An ensemble of low-bias decorrelated learners, combined with randomized inputs and prediction averaging, generally mitigates overfitting. Neural network ensembles have been widely used in the past for both regression and classification problems [Hansen & Salamon, 1990], [Giacinto & Roli, 2001], [Li et al., 2018]. However, state-of-the-art classification pipelines rarely make use of ensembling techniques due to the large dimensions of modern CNNs. Such an approach would become prohibitive from a computational point of view. On the other side, for domains in which training data is not abundant, ensembles of relatively small neural networks become highly attractive thanks to the aforementioned factors. Furthermore, ensemble approaches have also been shown to be good estimators of predictive uncertainty, an important tool for any predictive system [Lakshminarayanan et al., 2017].

Motivated by the nice variance-reduction properties of ensembles, we systematically study their effectiveness in small data tasks by a) fixing a computational budget and b) comparing them to corresponding deeper and wider single variants. Furthermore, we studied the performance of models varying depth, width, and ensembles dimension. According to our empirical study, ensembles are preferable over wider networks that are in turn better than deeper ones. The obtained results confirm our intuition and show that running a large ensemble is advantageous in terms of accuracy in domains with small data.

Finally, we study the effectiveness of two losses: 1) the widely used cross-entropy and 2) the recently proposed cosine loss [Barz & Denzler, 2020]. Despite the latter loss has been specifically proposed to tackle small data problems, we have noticed some cases in which the cross-entropy still gives higher accuracy. The combined factors of model complexity and the amount of available data seem to influence the outcome.

In summary, the contributions of our work are the following: *i*) we systematically study the use of neural ensembles in the small sample domain and show that they improve the state of the art; *ii*) we show that ensembles of smaller-scale networks outperform their computationally equivalent single competitors with increased depth and width; *iii*) we compare state-of-the-art losses showing that their performance depends on diverse factors and we provide a way of choosing the right configuration depending on the situation.

## 2 RELATED WORK

In this section, we first present a summary of the main techniques proposed in the literature to learn from a small sample. Secondly, we give an overview of neural ensembling techniques.

**Learning from a small sample** is extremely challenging and, for this reason, largely unsolved. As previously said, few works have tried to tackle the problem of training DL architectures with a small number of samples due to its difficulty.
We start by mentioning a series of works that focused on the classification of vector data and mainly used the UCI Machine Learning Repository as a benchmark. In [Fernández-Delgado et al., 2014] the authors have shown the superiority of random forests over a large set of classifiers including feed-forward networks. Later, [Olson et al., 2018] used a linear program to empirically decompose fitted neural networks into ensembles of low-bias sub-networks. They showed that these sub-networks were relatively uncorrelated which lead to an internal regularization process similar to what happens in random forests, obtaining comparable results. More recently, [Arora et al., 2020] proposed the use of neural tangent kernel (NTK) architectures in low data tasks and obtained significant improvements over all previously mentioned classifiers.
All previous works did not test CNNs since inputs were not images. In the computer vision domain, a straightforward approach to improve generalization is to implement techniques that try to synthesize new images through different transformations (e.g. data augmentation [Shorten & Khoshgoftaar, 2019]). Some previous knowledge regarding the problem at hand might turn to be useful in some cases [Hu et al., 2017]. However, this makes data augmentation techniques not always generalizable to all possible image classification domains. It has also been proposed to train generative models (e.g. GANs) to increase the dataset size and consequently, performance [Liu et al., 2019]. Generating new images to improve performance is extremely attractive and effective. Yet, training a generative model

might be computationally intensive or present severe challenges in the small sample domain. In our work, we mainly use standard data augmentation and do not focus on approaches that improve image synthesis. Indeed, we assume that any state-of-art data augmentation/generation technique could be coupled with our trained networks to further improve testing accuracy.

[Rueda-Plata et al., 2015] suggested to train CNNs with a greedy layer-wise method, analogous to that used in unsupervised deep networks and showed that their method could learn more interpretable and cleaner visual features. [Bietti et al., 2019] presented diverse kernel-based regularization techniques to regularize deep networks. [Barz & Denzler, 2020] proposed the use of the cosine loss to prevent overfitting claiming that the L2 normalization involved in the cosine loss is a strong, hyper-parameter-free regularizer when the availability of samples is scarce. On the other hand, [Arora et al., 2020] performed experiments with convolutional neural tangent kernel (CNTK) networks on small CIFAR-10 showing the superiority of CNTK in comparison to a ResNet-34 model. Finally, [Bornschein et al., 2020] studied the generalization performance of deep networks as the size of the training set varies. They found out that even larger networks can handle overfitting and obtain comparable or better results than smaller nets if properly optimized and calibrated.

**Ensembles of neural networks** are widely successful for improving both the accuracy and predictive uncertainty of very different predictive tasks, ranging from supervised learning [Hansen & Salamon, 1990] and few-shot learning [Dvornik et al., 2019], to model-based reinforcement learning [Deisenroth & Rasmussen, 2011]. The operational pipeline of neural ensembles consists of training multiple networks individually and then averaging their predictions obtaining, in the end, better results than single estimates [Opitz & Maclin, 1999]. The main reason behind this success is caused by the diversity of the solution found by each independent optimization process. Although ensemble members usually score similar testing performance, given the same input, they may disagree on the prediction, since they converged to different local minima [Garipov et al., 2018]. By combing the predictions, they benefit from diversity and decorrelated errors [Bishop, 1994]. Furthermore, similar to Bayesian methods [Gal & Ghahramani, 2016], they can estimate predictive uncertainty and provide more calibrated prediction probabilities [Lakshminarayanan et al., 2017]. For this reason, they can score the best performance on out-of-distribution uncertainty benchmarks. The main drawback of standard ensembles regards their computational cost that generally grows linearly with the ensemble dimension. Some works have tried to mitigate such problem in the field of continual and supervised learning [Wen et al., 2020], [Wasay et al., 2020].

However, since in this work we are running neural ensembles in low data regimes, the training time is relatively low. Neural ensembling has also been occasionally applied to specific problems and applications with small data [Veni & Rani, 2014], [Shaikhina & Khovanova, 2017], [Phung et al., 2019], [Han & Jeong, 2020].

**Our article contributes** to the field of small data and deep ensembles by proposing a structured study comparing both state-of-the-art models and different configurations of ensembles. Consistent results on different sub-sampled versions of popular datasets confirm the empirical validity of our study. Regarding deep ensembles with small data, this paper is the first one to perform a comprehensive analysis and detailed experimental evaluation. The experimental results of our paper will be useful for practitioners that are facing classification problems with relatively tiny datasets.

## 3 Deep ensembles

We are dealing with a problem in which we assume to have a dataset of samples and corresponding labels $\mathcal{D} = \{(\mathbf{x}_1, \mathbf{y}_1) \ldots, (\mathbf{x}_s, \mathbf{y}_s)\}$ where each label $\mathbf{y}$ is a one-hot vector that is encoding $K$ different classes. Furthermore, we are assuming that $\mathcal{D}$ is balanced and that the number of samples per class is equal to $N$ and relatively small.[1] In general, $\mathbf{x}$ might be a vector of any size. In our work, we are tackling a classification problem where each input sample is an image, hence, $\mathbf{x} \in \mathbb{R}^{H \times W \times D}$. The objective is to learn the input-output mapping $\mathbf{y} = f_\theta(\mathbf{x})$ with $f(\cdot)$ representing a function approximator and $\theta$ its parameters.

More precisely, since we are implementing a weighted average ensemble, we define a set $\mathcal{M} = \{g_{\theta_m}(\cdot) : m = 1, \ldots M\}$ of $M$ functions. In our particular implementation, the ensemble is homogeneous, therefore all functions $g$ are equal. Note that during our analysis, we will evaluate

---

[1]It is difficult to quantify a small dataset, because it is not an objective measure, but depends on multiple factors. More details are given in Section 4.1.

different ensembles varying the complexity of the base members $g$. In other words, we can define $\mathcal{M}$ as a function of $g$ respecting the constraint $\mathcal{M}(g^{(i)}) \sim \mathcal{M}(g^{(j)})$ for any $g^{(i)}$ and $g^{(j)}$ in terms of computational cost. At testing time, given an input sample $\mathbf{x}$, the ensemble prediction is going to be the expected value of the predictions coming from each member in $\mathcal{M}$ scaled by a function $\phi$ depending on the type of the loss used. More precisely, $\mathbf{y} = f_\theta(\mathbf{x}) = \frac{1}{M} \sum_{m=1}^{M} \phi\left(g_{\theta_m}(\mathbf{x})\right)$. The weights $\theta_m$ are randomly initialized and ensemble members update their parameters receiving the same input batch. Throughout the optimization process, the members are independent, therefore each network will minimize its own loss $\mathcal{L}(\mathbf{y}, g_{\theta_m}(\mathbf{x}))$.

In this work, we implement two loss functions. The first one is the popular categorical cross-entropy which is computed as $\mathcal{L}_{xe} = -\mathbf{y} \cdot \log\left(\frac{\exp(g_{\theta_m}(\mathbf{x}))}{||\exp(g_{\theta_m}(\mathbf{x}))||_1}\right)$ assuming that the exponential and logarithm are applied element-wise. In this case, $\phi$ corresponds to the soft-max.

The second one is the cosine loss, that was lately proposed to improve the generalization of deep networks with small datasets [Barz & Denzler, 2020]. The cosine loss simply computes the dot product among the one-hot encoded categories and the L2 normalized predictions as $\mathcal{L}_{cosine} = \mathbf{y} \cdot \frac{g_{\theta_m}(\mathbf{x})}{||g_{\theta_m}(\mathbf{x})||_2}$. In this other case, $\phi$ corresponds to the L2 normalization. The improvements reported in [Barz & Denzler, 2020] were explained by the L2 regularization that prevents the network from focusing on the magnitudes and moves the attention to vector directions. Different studies have shown that directions maintain more information in feature space [Husain & Bober, 2016], [Zhe et al., 2019]. Moreover, differently from the cosine loss, the cross-entropy is steeper, not symmetric, and unbounded. These three characteristics might be indeed problematic when facing very small datasets since noisy gradient updates may cause instability.

To sum up, we are going to train multiple deep ensembles varying the complexity of their base members but keeping the overall computational cost fixed. The networks in the groups are homogeneous and trained with the previously cited losses.

## 4 EXPERIMENTAL PROCEDURE

In this section, we describe the experimental procedure of our work. We provide a comprehensive evaluation exploring multiple directions including different datasets, architectures, and layouts of the networks.

### 4.1 DATASETS

We perform our study on four popular benchmarks for image classification: CIFAR-10, CIFAR-100 [Krizhevsky et al., 2009], SVHN [Netzer et al., 2011] and Stanford Dogs [Khosla et al., 2011].

CIFAR-10 is an established computer vision benchmark consisting of color images coming from 10 different classes of objects and animals. The dataset has originally $50,000$ training images and $10,000$ testing images with both sets balanced. The input images are of size $32 \times 32 \times 3$.

CIFAR-100 is the more complex version of CIFAR-10 being composed of 100 classes and containing the same number of training and testing samples. The training/testing splits and input dimensionalities are equal to the ones of CIFAR-10.

SVHN is a real-world image dataset semantically similar to MNIST since contains images of digits. The popular cropped-version sets used originally have $73,257$ training and $26,032$ testing images of dimension $32 \times 32 \times 3$.

Finally, Stanford Dogs is a fine-grained categorization problem consisting of 120 species of dogs. The dataset has been built using samples and annotation from ImageNet and contains images with sides not smaller than 200 pixels. The original training set contains 100 samples per class while the test set has in total $8,580$ images.

Sub-sampling the original training sets is necessary since we are interested in benchmarking the capabilities of our ensembles when the sample size is limited. To this end, we sub-sample the training sets of the chosen datasets to match our specifications. More in detail, the number of samples per class is varied in the set $\{10, 50, 100, 250\}$. For Stanford Dogs, we stop at 100 samples having reached the original size of the training dataset. A similar procedure was already proposed in [Barz & Denzler, 2020].

Note that the test datasets are not changed, and remain fixed throughout all the evaluations. To ensure consistency of the results, we perform 5 runs for each sub-sampled version of each dataset.

## 4.2 COMPARED MODELS

To build our ensembles we considered three different families of convolutional architectures: VGG [Simonyan & Zisserman, 2014], ResNet [He et al., 2016] and DenseNet [Huang et al., 2017]. We tested the first two on the CIFAR datasets, while DensNet on SVHN and Stanford Dogs.

We decided to measure the models' computational complexity as the number of floating-point operations (FLOPs) to make a prediction. This is a standard metric that measures algorithmic/model complexity.[2] Note that when we have two networks of the same family and layout, a similar number of FLOPs also corresponds to a similar number of trainable parameters. We designed deeper and wider models keeping the same computational budget, having, therefore, an equivalent number of trainable parameters.

Different computational budgets were set depending on the dataset and network architecture. For ResNet and VGG on CIFAR, the maximum computational budget was, respectively, set to be roughly close to the FLOPs of a single ResNet-110 and VGG-9. For DenseNet on SVHN, we have chosen the complexity of DenseNet-BC-52 while, for Stanford Dogs, DenseNet-BC-121.

We compare the chosen deepest network with: a) an ensemble of shallower networks and b) a shallower network with increased width, matching, in any case, the available computational budget. More in detail, we train: 1) an ensemble of 20 ResNet-8 and a single ResNet-8 with 72 base filters; 2) an ensemble of 5 VGG-5 with 32 base filters and a single VGG-5 with 76 filters; 3) an ensemble of 6 DenseNet-BC-16 and a single DenseNet-BC-16 with growth rate ($k$) equal to 30; 4) an ensemble of 3 DenseNet-BC-62 and a single DenseNet-BC-62 with $k$= 56.

Moreover, to study the influence of depth, width, and ensemble dimension, we also set two smaller computational budgets for ResNets on the CIFAR datasets matching the FLOPs of a single ResNet-26 and ResNet-50 with 16 base filters. We kept the ResNet-8 architecture to make the corresponding wider and ensemble configuration. More in detail, a single ResNet-8 with 36/50 base filters and 5/10 ResNet-8.

Finally, we compare our ResNet ensembles trained with the two losses evaluating the 20 ResNet-8 ensembles, a smaller ensemble of 5 ResNet-20, and the deepest ResNet-110. In general, if not specified, networks are trained with the cross-entropy loss.

For more details regarding the computational complexity, structure and training configurations refer to Appendices A and B.

## 5 RESULTS

### 5.1 COMPARISON WITH THE STATE OF THE ART

First, we show the effectiveness of our ensembles comparing them with state-of-the-art techniques that were proposed for learning from a small sample. More precisely, diverse kernel-based regularizations proposed in [Bietti et al., 2019], and the CNTK tested in [Arora et al., 2020]. Both of these works have chosen to test their models on sub-sampled versions of CIFAR-10. In this section, our experimental procedure is adapted to match the experiments of the original papers.

The first work by [Bietti et al., 2019] evaluates their models with and without standard augmentation composed of random cropping and mirroring. Their training sets were comprised of 100 and 500 samples per class. For sake of completeness, we also evaluate our ensembles in the latter case, despite the training set already contains a consistent number of samples. They trained a VGG-11 and ResNet-18 with various regularization techniques. To keep a fair comparison, we directly compared an ensemble of 10 VGG-5 and 20 ResNet-8 with the corresponding regularized architectures tested in [Bietti et al., 2019]. Note, however, that with the default widths, both VGG-11 and ResNet-18 have many more training parameters than our ensembles and that in the original work, authors tuned the hyper-parameters of their methods on a held-out validation set.[3]

---

[2]In our implementation, we used Tensorflow APIs to compute the FLOPs of networks.

[3]This is a simplifying assumption that inevitably increases the dimension of the training set.

On the other hand, authors in [Arora et al., 2020] used a very restrictive training protocol varying the number of samples per class from 1 to 128 and not using any kind of data augmentation. Here, we compare our ensemble of 20 ResNet-8 with their baseline ResNet-34 and proposed CNTK. Also, in this case, our ensemble has fewer total trainable parameters than a single ResNet-34.

The results are shown in Table 1. In all cases, our ensembles outperform the competing techniques. These results clearly show that standard ensembling is a simple yet effective way to reduce model variance and consequently, increase performance with few training samples.

Table 1: Comparison of our ensembles with state-of-the-art models that were proposed in [Bietti et al., 2019] and [Arora et al., 2020], and evaluated on sub-sampled versions of CIFAR-10.

(a) We follow the evaluation protocol with/without data augmentation and 100/500 samples per class. We report the results of our CNN-4 and ResNet-8 ensembles. For each pairwise comparison, we highlight the highest value in bold font. † Best of Table 4 from [Bietti et al., 2019].

| Model | Aug | N = 100 | Model | Aug | N = 500 |
|---|---|---|---|---|---|
| VGG-11 + grad-$l_2$ + SN proj[†] | ✗ | 46.88 | VGG-11 + PGD-$l_2$ + SN proj[†] | ✗ | 64.50 |
| 10 VGG-5 | ✗ | **52.16 $\pm$ 0.38** | 10 VGG-5 | ✗ | **68.59 $\pm$ 0.21** |
| ResNet-18 + $\|\|\nabla f\|\|^2$[†] | ✗ | 44.97 | ResNet-18 + $\|\|f\|\|_\delta^2$ SN proj[†] | ✗ | 59.03 |
| 20 ResNet-8 | ✗ | **52.66 $\pm$ 0.45** | 20 ResNet-8 | ✗ | **69.49 $\pm$ 0.36** |
| VGG-11 + grad-$l_2$ + SN proj[†] | ✓ | 55.32 | VGG-11 + grad-$l_2$[†] | ✓ | 75.38 |
| 10 VGG-5 | ✓ | **57.73 $\pm$ 0.52** | 10 VGG-5 | ✓ | **76.26 $\pm$ 0.09** |
| ResNet-18 + grad-$l_2$[†] | ✓ | 49.30 | ResNet-18 grad-$l_2$ + SN proj[†] | ✓ | 77.73 |
| 20 ResNet-8 | ✓ | **57.38 $\pm$1.5** | 20 ResNet-8 | ✓ | **81.78 $\pm$ 0.19** |

(b) We evaluate 20 ResNet-8 following the set-up of [Arora et al., 2020] that does not include data augmentation. The number of samples per class ranges from 1 to 128. For each sub-sampled dataset, we highlight the highest value in bold font. † Best of Table 2 from [Arora et al., 2020].

| Model | N = 1 | N = 2 | N = 4 | N = 8 | N = 16 | N = 32 | N = 64 | N = 128 |
|---|---|---|---|---|---|---|---|---|
| ResNet-34[†] | 14.59 $\pm$ 1.99 | 17.5 $\pm$ 2.47 | 19.52 $\pm$ 1.39 | 23.32 $\pm$ 1.61 | 28.3 $\pm$ 1.38 | 33.15 $\pm$ 1.2 | 41.66 $\pm$ 1.09 | 49.14 $\pm$ 1.31 |
| CNTK[†] | 15.33 $\pm$ 2.43 | 18.79 $\pm$ 2.13 | 21.34 $\pm$ 1.91 | 25.48 $\pm$ 1.91 | 30.48 $\pm$ 1.17 | 36.57 $\pm$ 0.88 | 42.63 $\pm$ 0.68 | 48.86 $\pm$ 0.68 |
| 20 ResNet-8 | **16.73 $\pm$ 1.02** | **20.66 $\pm$ 0.53** | **24.47 $\pm$ 0.9** | **29.54 $\pm$ 0.91** | **34.55 $\pm$ 0.92** | **40.48 $\pm$ 1.28** | **47.5 $\pm$ 0.24** | **55.04 $\pm$ 0.46** |

## 5.2 VARIATION OF DATASETS AND ARCHITECTURES

In this section, we report the results of the ensembles and analyze their relation with their deeper and wider single competitors. From Table 2, we note that ensembles are the best option in terms of testing accuracy for all cases except two (SVHN with 10 and CIFAR-100 with 250 samples per class). All models are trained with standard data augmentation, for more details refer to Appendix B.

We notice that deeper networks struggle with small datasets. On CIFAR benchmarks, ResNet-8 and VGG-5 ensembles obtain very large gains over deeper architectures ranging from a minimum of $\sim 3\%$ to a maximum of $\sim 15\%$. To be sure that these results are not just due to a poor regularization of the deepest model, we also make a test with more aggressive data augmentation available in Table 4 of Appendix C. However, results remained consistent with the ones with standard augmentation. Considering DenseNets, we appreciate a similar trend, with more moderate but still significant gains, from $\sim 1.5\%$ to $\sim 6\%$.

On the other hand, wider networks seem to handle the lack of training data better than deeper models. Still, model averaging and independent training has a clear advantage over wider single networks. ResNet, VGG and DenseNet ensembles generalize better than wider nets in terms of testing accuracy from a minimum of $\sim 1\%$ to a maximum of $\sim 5\%$. Additional experiments and discussions regarding depth width and ensembles will follow in the next sections.

## 5.3 VARIATION OF DEPTH, WIDTH AND SIZE OF THE ENSEMBLE

Here, we show more experiments with ResNets on the CIFAR datasets to understand how the different factors that we considered to build our ensembles and direct competitors affect the results. In other

Table 2: Comparison between deep networks, wide networks and ensembles of less complex models. $M$ indicates the number of networks in the ensemble and $N$ the number of training samples per class. All networks are trained with standard data augmentation, details are available in Appendix B. Means and standard deviations are obtained from five independent runs.

(a) CIFAR-10

| Model | M | N = 10 | N = 50 | N = 100 | N = 250 |
|---|---|---|---|---|---|
| ResNet-110-16 | 1 | $26.06 \pm 0.56$ | $41.32 \pm 0.58$ | $49.21 \pm 1.04$ | $62.5 \pm 1.49$ |
| ResNet-8-72 | 1 | $29.65 \pm 1.54$ | $48.0 \pm 0.72$ | $58.16 \pm 0.37$ | $72.41 \pm 0.36$ |
| ResNet-8-16 | 20 | $\mathbf{32.83 \pm 2.39}$ | $\mathbf{52.88 \pm 0.92}$ | $\mathbf{63.64 \pm 0.61}$ | $\mathbf{76.23 \pm 0.28}$ |
| VGG-9-32 | 1 | $27.64 \pm 1.28$ | $41.74 \pm 0.11$ | $47.22 \pm 0.42$ | $56.36 \pm 1.52$ |
| VGG-5-76 | 1 | $30.28 \pm 1.37$ | $45.39 \pm 0.56$ | $51.38 \pm 0.72$ | $62.08 \pm 1.16$ |
| VGG-5-32 | 5 | $\mathbf{31.69 \pm 1.03}$ | $\mathbf{48.61 \pm 0.74}$ | $\mathbf{57.18 \pm 0.61}$ | $\mathbf{68.38 \pm 0.47}$ |

(b) CIFAR-100

| Model | M | N = 10 | N = 50 | N = 100 | N = 250 |
|---|---|---|---|---|---|
| ResNet-110-16 | 1 | $8.62 \pm 1.79$ | $29.44 \pm 0.5$ | $40.84 \pm 0.41$ | $60.98 \pm 1.8$ |
| ResNet-8-72 | 1 | $16.51 \pm 0.38$ | $42.52 \pm 0.44$ | $54.94 \pm 0.8$ | $\mathbf{66.38 \pm 0.12}$ |
| ResNet-8-16 | 20 | $\mathbf{18.92 \pm 0.38}$ | $\mathbf{46.56 \pm 0.41}$ | $\mathbf{57.37 \pm 0.05}$ | $65.56 \pm 0.21$ |
| VGG-9-32 | 1 | $10.22 \pm 0.38$ | $23.94 \pm 0.34$ | $31.04 \pm 0.59$ | $42.09 \pm 1.01$ |
| VGG-5-76 | 1 | $13.25 \pm 0.07$ | $26.46 \pm 0.36$ | $33.52 \pm 0.39$ | $44.84 \pm 0.67$ |
| VGG-5-32 | 5 | $\mathbf{16.29 \pm 0.57}$ | $\mathbf{34.37 \pm 0.33}$ | $\mathbf{44.04 \pm 0.17}$ | $\mathbf{56.37 \pm 0.05}$ |

(c) SVHN

| Model | M | N = 10 | N = 50 | N = 100 | N = 250 |
|---|---|---|---|---|---|
| DenseNet-BC-52, k=12 | 1 | $\mathbf{16.72 \pm 1.75}$ | $78.42 \pm 1.19$ | $86.52 \pm 0.24$ | $89.6 \pm 0.7$ |
| DenseNet-BC-16, k=30 | 1 | $16.44 \pm 3.8$ | $76.41 \pm 1.65$ | $85.41 \pm 0.52$ | $89.28 \pm 0.06$ |
| DenseNet-BC-16, k=12 | 6 | $14.01 \pm 2.5$ | $\mathbf{82.02 \pm 1.67}$ | $\mathbf{87.73 \pm 0.44}$ | $\mathbf{91.61 \pm 0.32}$ |

(d) Stanford Dogs

| Model | M | N = 10 | N = 50 | N = 100 |
|---|---|---|---|---|
| DenseNet-BC-121, k=32 | 1 | $6.93 \pm 0.86$ | $28.32 \pm 1.33$ | $47.7 \pm 1.17$ |
| DenseNet-BC-62, k=56 | 1 | $7.33 \pm 0.35$ | $29.25 \pm 0.76$ | $47.82 \pm 0.83$ |
| DenseNet-BC-62, k=32 | 3 | $\mathbf{8.42 \pm 0.02}$ | $\mathbf{35.12 \pm 0.68}$ | $\mathbf{53.39 \pm 0.45}$ |

words, we aim at understanding what is the best direction, among depth, width, and number of networks in an ensemble, for allocating computation. To provide a complete picture, we set, as mentioned in Section 4.2, additional computational budgets in terms of FLOPs that are lower than the FLOPs of a single ResNet-110. The results of this analysis are visible in Figure 1. In the plots, the four budgets are represented by the different colors assuming green as the lowest budget (single ResNet-8-16) and black as the highest budget (single ResNet-110-16). Therefore, an increment of testing accuracy from green to black indicates a favorable allocation of resources.

A first result is that increasing the computational budget in the depth direction is not beneficial for small datasets (left plots in Figure 1). Indeed, we note that the deepest model is always the worst, except for the case of CIFAR-100 and 250 samples per class. This is reasonable since it is known that deep networks coupled with larger datasets are able to learn very complex functions that translate in good representational capacity [Montúfar et al., 2014].

Investing on network's width is slightly better (middle plots in Figure 1). On CIFAR-100, wider networks outperform the baseline model. This result is expected, since, on a more complex and larger problem, more capacity gives an advantage to the wider network. On the other hand, on CIFAR-10, the wider models perform comparably well to ResNet-8-16.

Finally, we note that ensembles have a clear advantage over the single ResNet-8-16 on both CIFAR-10 and CIFAR-100, with larger gains obtained on the latter (right plots in Figure 1). This makes ensembles the most preferable approach having the possibility of increasing the computational budget when facing a small dataset. We also note that the largest gap is generally between the single network and 5 ResNet-8 proving that the performance gain is already considerable with a relatively small ensemble.

## 5.4 WHY ARE ENSEMBLES MORE SAMPLE-EFFICIENT?

To better understand why ensembles perform better than competitors with small datasets, we focus on the results of both deeper and wider networks presented in previous sections.

We designed deeper and wider models of the same family keeping the same computational budget, having, therefore, an equivalent number of trainable parameters. Reasonably, both networks should be similar in terms of representational capacity and, in turn, generalization. Yet, deeper networks score the worst results in almost all cases (except DenseNets on SVHN). These findings are in line with the model proposed in [Geiger et al., 2019] that studied the optimization of feed-forward neural networks as a constraint satisfaction problem with continuous degrees of freedom. In agreement with their model, networks in the over-parametrized regime easily find good local minima. However, over-parametrization is a function of the *effective* number of parameters, i.e., all the parameters that affect the output function. According to the above-cited study, it is possible that, because of bad signal propagation inside the network (e.g. many ReLU activations not active) or parameters initialization, a network could have a smaller number of *effective* parameters and consequently struggles to find a good local minimum.

Therefore, taking this argument and our results into account, it is reasonable to believe that small datasets limit the expressive potential of deeper networks due to optimization challenges stemming from the depth and lack of data. Wider networks, thanks to their shallower structure, can better handle such difficulties. Yet, as also theorized and reported in [Geiger et al., 2020], ensembles of over-parametrized networks generalize better than larger models thanks to reduced model variance resulting from model averaging and independent training. Our results align with the results shown in [Geiger et al., 2020] on vanilla problems with large datasets and extend them to the domain of small datasets.

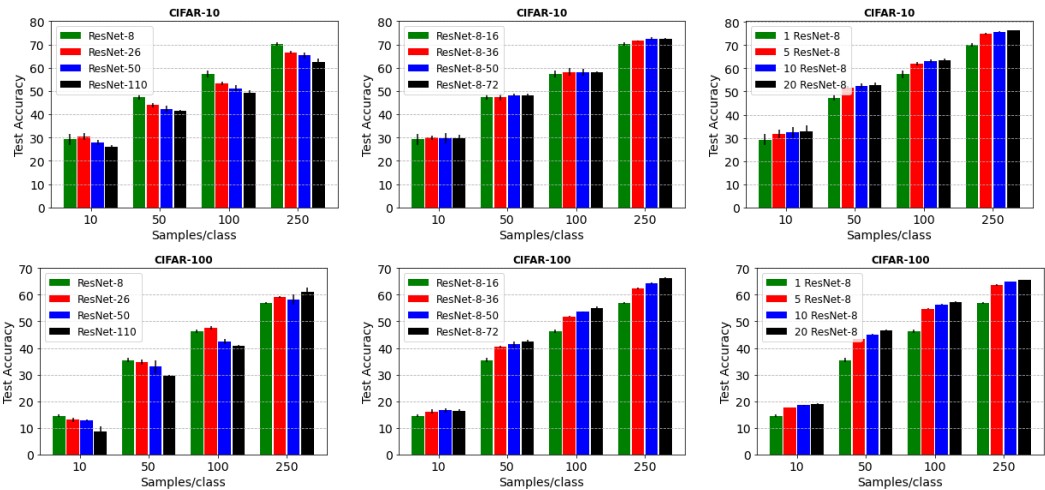

Figure 1: Results considering, respectively, the variation of depth, width and number of networks in the ensemble for ResNets on CIFAR datasets.

## 5.5 VARIATION OF THE LOSSES

In this final section, we analyze the effectiveness of the two losses evaluating them on the three ResNet models mentioned in Section 4.2. We note that considerations on the influence of losses on ensemble approaches are directly transferable to single network cases. For this reason, we show the comparison between single networks in Appendix C.

From the plots, we observe that model complexity and problem dimension seem to play a crucial role in the loss choice. Indeed, for the least complex ResNet-8, the cross-entropy is slightly better than the cosine loss on CIFAR-10 and substantially better on CIFAR-100. If we increase model complexity and analyze the ResNet-20 results, we note that the cosine loss outperforms the cross-entropy on CIFAR-10 but the opposite result is obtained on CIFAR-100. Finally, the cosine loss is clearly more convenient with ResNet-110 on CIFAR-10 and CIFAR-100 up to 100 samples per class. On the other

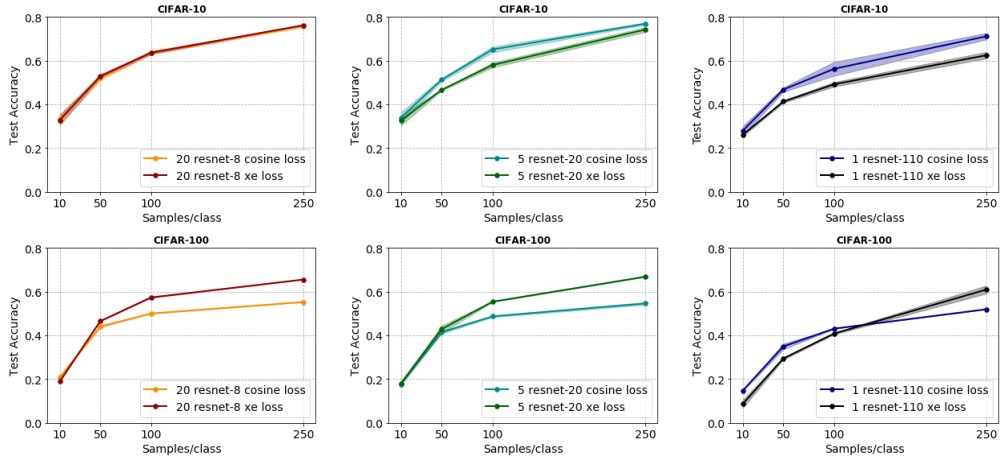

Figure 2: Comparison between the two losses considering the 20 ResNet-8, 5 ResNet-20 ensembles and a single ResNet-110.

hand, when training on CIFAR-100 with $N = 250$, the cross-entropy increases its effectiveness. The performance of ResNet-110 on CIFAR-100 obtained here follows the same behavior that was shown in Figure 2 of [Barz & Denzler, 2020]. Although results are not directly comparable because of a different training set-up, their cross-entropy network outperformed the cosine one after 200 samples per class.[4]

To summarize the results of this analysis, the cosine loss is more convenient if the total amount of data is scarce and the complexity of the network is medium-high. On the other hand, the cross-entropy is better with larger training sets and with medium-low network complexities.[5] We find these results in agreement with our previous argument regarding the difficulty of training deeper models on small datasets. Possibly, the bounded and smoother cosine loss provides a simpler optimization landscape that allows deeper networks to better propagate the signal and *effectively* exploit their potential using more parameters. On the other hand, with simpler architectures and/or more samples, depending on the specific problem, the steeper cross-entropy has an advantage over the cosine.

## 6 CONCLUSIONS

In this paper, we have investigated the capability of deep ensembles on image classification problems with limited data. Through comparing ensembles given a fixed computational budget, our results show that ensembles: a) beat current state-of-the-art approaches for learning from small datasets; b) perform generally better than their deeper and wider single competitors.

We would like to stress that these trends are consistent on all the four tested datasets and architectures. Such encouraging results suggest the potential of using ensembles for small datasets and further developments of this line of research in the field of small data.

Moreover, we studied the performance of models varying depth, width, and ensembles dimension. We provided empirical evidence that theories lately proposed in [Geiger et al., 2019] and [Geiger et al., 2020] seem to fit well to our particular settings with small data. Consequently, they might turn out to be useful in future studies.

Finally, we provide an answer to the question of whether the cosine loss has a clear advantage over the cross-entropy. From our results, we partially confirm the nice regularization capabilities of the cosine loss, but we also broaden the picture showing that the availability of data and the complexity of the model are two important factors to consider before making a choice.

We believe that the results reported here can be exploited by the community both as baselines for future woks and as a practical guide for solving similar problems.

---

[4]They trained ResNet-110 with twice as many channels per-layer, used SGD with Warm Restarts, more aggressive data augmentation, and fine-tuned the learning rates.

[5]Fixing the number of samples per class causes sub-sampled CIFAR-100 to be ten times larger than sub-sampled CIFAR-10.

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

## A  ADDITIONAL DETAILS ON THE ARCHITECTURES

In this section, we provide more specifications about the used architectures and their computational complexities.

For what concerns ResNets and DenseNets, we used the original structures proposed, respectively, for datasets of smaller images (e.g. CIFAR-10) in [He et al., 2016] and [Huang et al., 2017]. Therefore, if not otherwise specified, the default base width of ResNets is equal to 16 and the growth rate of DenseNets is set to 12. Since Stanford Dogs has images of larger sizes, we have chosen a larger architecture that was trained in the original paper on ImageNet. To make a smaller competitor and build an ensemble, we halved the number of dense blocks in each stack of DenseNet-BC-121 making a DenseNet-BC-52. In this case, the default growth rate is equal to 32.
Finally, we removed the two densely connected layers at the bottom of VGG-11 to make a less computationally expensive VGG-9. Moreover, we set the base width to 32 filters instead of 64 to further decrease the computational load. The shortest version, that we called VGG-5, is obtained by keeping a single convolutional layer for the first four original blocks. Before the last dense layers, all VGG architectures are equipped with a dropout layer with drop-rate equal to 0.4.

In Table 3 we report the computational complexity of single networks used in the corresponding dataset. To obtain the computational complexity of ensembles, directly multiply the FLOPs of a single network by the number of total networks in the ensemble.

| Model | CIFAR-10 | CIFAR-100 | SVHN | Stanford Dogs |
|---|---|---|---|---|
| ResNet-8-16 | 0.46 | 0.47 | – | – |
| ResNet-8-36 | 2.33 | 2.35 | – | – |
| ResNet-8-50 | 4.48 | 4.52 | – | – |
| ResNet-8-72 | 9.29 | 9.34 | – | – |
| ResNet-20-16 | 1.62 | 1.63 | – | – |
| ResNet-26-16 | 2.21 | 2.22 | – | – |
| ResNet-50-16 | 4.54 | 4.55 | – | – |
| ResNet-110-16 | 10.36 | 10.37 | – | – |
| VGG-5-32 | 0.79 | 0.98 | – | – |
| VGG-5-76 | 4.41 | 4.85 | – | – |
| VGG-9-32 | 4.61 | 4.66 | – | – |
| DenseNet-BC-16, k=12 | – | – | 0.091 | – |
| DenseNet-BC-16, k=30 | – | – | 0.54 | – |
| DenseNet-BC-52, k=12 | – | – | 0.54 | – |
| DenseNet-BC-62, k=32 | – | – | – | 4.79 |
| DenseNet-BC-62, k=56 | – | – | – | 14.36 |
| DenseNet-BC-121, k=32 | – | – | – | 14.32 |

Table 3: Computational complexity of single networks in terms of MFLOPs.

## B  ADDITIONAL DETAILS ON TRAINING SCHEDULES

In this section, we provide more details regarding the training set-up used to train our models.

**Data pre-processing**   In all experiments, input images are normalized subtracting the per-channel mean computed over the sub-sampled training set.
On CIFAR datasets we adopted standard data augmentation widely used in previous works [He et al., 2016], [Huang et al., 2017]. More precisely, each image is padded 4 pixels on each side and a $32 \times 32$ crop is randomly sampled from the padded image or its horizontal flip.
For the evaluation with more aggressive data augmentation shown in Table 4 we added color distortion including random hue, contrast, brightness and saturation. We further used pixel-level random erasing with the default parameters proposed in the original paper for CIFAR-10 [Zhong et al., 2020].
On SVHN, we use the same cropping and flipping strategy and we add random contrast and brightness.

Finally, on Stanford Dogs, all input images were resized to $256 \times 256$ pixels. Before being fed to the network, we used random crop and flipping of $224 \times 224$ patches. Further, images were color distorted again with random hue, contrast, brightness and saturation. At evaluation time, we apply a single central crop of size $224 \times 224$.

**Optimization** For training ResNets and DenseNets, we use stochastic gradient descent (SGD) with weight decay and Nesterov momentum respectively set to $10^{-4}$ and $0.9$ as was done in the original papers [He et al., 2016] and [Huang et al., 2017]. We start with a learning rate of $0.1$ and decrease it after 75% of the total number of iterations by an order of magnitude. We increased the number of iterations with the initial learning rate to be sure of decreasing it after having reached the training loss plateau. Furthermore, we have noticed that decreasing a second time the learning rate did not improve further the testing performance. For the deeper ResNet networks (i.e. ResNet-26, ResNet-50 and ResNet-110) we noticed some instability and larger variance in terms of testing accuracy on CIFAR-10. For this reason, in that case, we trained them with the starting learning rate equal to $0.01$ and then follow the same piece-wise schedule. We obtained more stable and faster convergence along with better results especially for the smaller datasets. On the other hand, VGG architectures were trained with Adam optimizer and default hyper-parameters.

All networks were fed with mini-batches of 32 images. We preferred a smaller batch with respect to the original 128 since led to better performance. Despite this effect is well-known in DL [Keskar et al., 2017], we empirically noticed that this is even more influential in our settings since, probably, more noisy gradient updates prevent the network from overfitting the tiny training sets.

All ResNets and DenseNets were trained for the same number of epochs. We noticed that VGG architectures needed less training epochs to converge (probably due to the faster convergence of Adam optimizer). Assuming that the first element of the set corresponds to datasets with 10 samples per class and the last one to 250, architectures are trained, respectively for $\{400, 300, 300, 250\}$ epochs.

## C ADDITIONAL EXPERIMENTS

Finally, we provide additional experiments mentioned in the paper.

First, we show the comparison between the losses in the case of single ResNet-8 and ResNet-20. As already mentioned, the reasoning made in Section 5.5 applies also to the single network case.

Then, in Table 4 we show an additional test that we run with stronger data augmentation.
We note that all networks obtain better performance with respect to the baseline augmentation. However, the advantage of ensembles over deeper networks is still wide, from a minimum of $\sim 5\%$ to a maximum of $\sim 13\%$. Further, ensembles still outperform wide networks by significant margins. We note, though, that this gap is reduced by a couple of percentage points. Thus, more aggressive augmentation has helped in a more significant way the widest model. Therefore, we reasonably expect that, in the limit of adding more data (i.e. not the scenario of our work) the wider model is going to match and eventually outperform the group of less complex networks.

Finally, in Table 5 we show a different view of Figure 3 to better understand the comparison between the ensembles and their competitors. It can be clearly seen that ResNet-8 ensembles outperform the competing wide ResNet-8 and deep ResNet at all three computational levels.

Table 4: Comparison between deep, wide and ensembles of ResNets on CIFAR-10 using more aggressive data augmentation (see Appendix B). Means and standard deviations are obtained from five independent runs.

| Model | M | N = 10 | N = 50 | N = 100 | N = 250 |
|---|---|---|---|---|---|
| ResNet-110-16 | 1 | $29.05 \pm 0.8$ | $45.77 \pm 0.64$ | $57.81 \pm 0.96$ | $67.47 \pm 1.21$ |
| ResNet-8-72 | 1 | $32.29 \pm 1.47$ | $55.54 \pm 0.94$ | $65.28 \pm 1.21$ | $75.61 \pm 0.62$ |
| ResNet-8-16 | 20 | $\mathbf{34.27 \pm 0.35}$ | $\mathbf{58.15 \pm 0.57}$ | $\mathbf{67.4 \pm 0.51}$ | $\mathbf{77.07 \pm 0.13}$ |

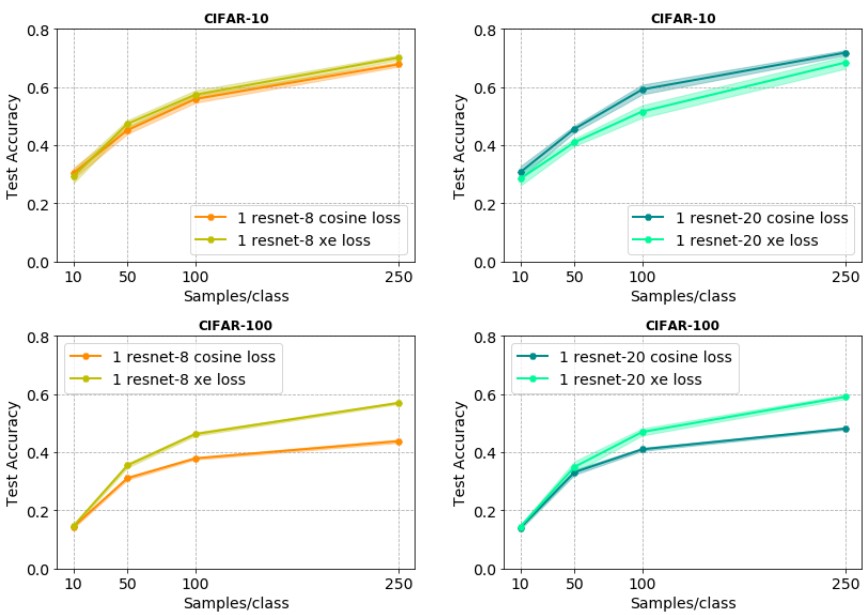

Figure 3: Comparison between the two losses considering single ResNet-8 and ResNet-20.

Table 5: This table provides a different view of Figure 3 reporting the exact means and standard deviations obtained on 5 independent runs. The comparison is made at different computational levels between deep, wide and ensembles using the chosen ResNet architectures on CIFAR-10 and CIFAR-100 datasets.

(a) CIFAR-10

| Model | M | N = 10 | N = 50 | N = 100 | N = 250 |
|---|---|---|---|---|---|
| ResNet-26-16 | 1 | $30.49 \pm 1.51$ | $43.91 \pm 0.79$ | $53.29 \pm 0.84$ | $66.68 \pm 0.42$ |
| ResNet-8-36 | 1 | $29.94 \pm 0.87$ | $47.49 \pm 1.42$ | $58.24 \pm 1.49$ | $71.5 \pm 0.35$ |
| ResNet-8-16 | 5 | $\mathbf{31.75 \pm 1.78}$ | $\mathbf{51.75 \pm 0.62}$ | $\mathbf{61.98 \pm 0.81}$ | $\mathbf{74.85 \pm 0.36}$ |
| ResNet-50-16 | 1 | $27.78 \pm 1.13$ | $42.27 \pm 1.62$ | $51.23 \pm 1.45$ | $65.34 \pm 1.36$ |
| ResNet-8-50 | 1 | $29.66 \pm 2.32$ | $48.13 \pm 0.55$ | $58.1 \pm 1.6$ | $72.54 \pm 0.67$ |
| ResNet-8-16 | 10 | $\mathbf{32.41 \pm 2.34}$ | $\mathbf{52.44 \pm 1.02}$ | $\mathbf{63.03 \pm 0.66}$ | $\mathbf{75.84 \pm 0.31}$ |
| ResNet-110-16 | 1 | $26.06 \pm 0.56$ | $41.32 \pm 0.58$ | $49.21 \pm 1.04$ | $62.5 \pm 1.49$ |
| ResNet-8-72 | 1 | $29.65 \pm 1.54$ | $48.0 \pm 0.72$ | $58.16 \pm 0.37$ | $72.41 \pm 0.36$ |
| ResNet-8-16 | 20 | $\mathbf{32.83 \pm 2.39}$ | $\mathbf{52.88 \pm 0.92}$ | $\mathbf{63.64 \pm 0.61}$ | $\mathbf{76.23 \pm 0.28}$ |

(b) CIFAR-100

| Model | M | N = 10 | N = 50 | N = 100 | N = 250 |
|---|---|---|---|---|---|
| ResNet-26-16 | 1 | $13.02 \pm 0.89$ | $34.71 \pm 0.97$ | $47.61 \pm 0.71$ | $59.07 \pm 0.38$ |
| ResNet-8-36 | 1 | $16.13 \pm 0.88$ | $40.45 \pm 0.24$ | $51.84 \pm 0.23$ | $62.37 \pm 0.46$ |
| ResNet-8-16 | 5 | $\mathbf{17.7 \pm 0.08}$ | $\mathbf{43.44 \pm 0.02}$ | $\mathbf{54.78 \pm 0.23}$ | $\mathbf{63.76 \pm 0.28}$ |
| ResNet-50-16 | 1 | $12.74 \pm 0.38$ | $33.05 \pm 2.44$ | $42.48 \pm 1.03$ | $58.19 \pm 1.87$ |
| ResNet-8-50 | 1 | $16.53 \pm 0.78$ | $41.36 \pm 1.2$ | $53.55 \pm 0.19$ | $64.37 \pm 0.41$ |
| ResNet-8-16 | 10 | $\mathbf{18.48 \pm 0.17}$ | $\mathbf{45.15 \pm 0.11}$ | $\mathbf{56.42 \pm 0.3}$ | $\mathbf{64.98 \pm 0.01}$ |
| ResNet-110-16 | 1 | $8.62 \pm 1.79$ | $29.44 \pm 0.5$ | $40.84 \pm 0.41$ | $60.98 \pm 1.8$ |
| ResNet-8-72 | 1 | $16.51 \pm 0.38$ | $42.52 \pm 0.44$ | $54.94 \pm 0.8$ | $\mathbf{66.38 \pm 0.12}$ |
| ResNet-8-16 | 20 | $\mathbf{18.92 \pm 0.38}$ | $\mathbf{46.56 \pm 0.41}$ | $\mathbf{57.37 \pm 0.05}$ | $65.56 \pm 0.21$ |

