# OpenReview forum: "On the Effectiveness of Deep Ensembles for Small Data Tasks"
_ICLR.cc/2021/Conference — Reject_

### Official Review · AnonReviewer2 · 2020-10-28
**needs more extensive evaluation and comparison with related methods**

**Rating:** 3
**Confidence:** 4

**Review:**

The paper studies the performance of ensembles of deep networks on small-data tasks taken from subsets of Cifar10 and Cifar100, with either the cross-entropy loss or the cosine loss. The authors conduct extensive experiments on these datasets with various choices of sample size, and are careful of evaluating models with comparable computational budget, by considering ResNet architectures of varying depths and with different numbers of models in the ensemble. They find that ensembles of small models tend to outperform single large models.

The approach seems promising, and the extensive experiments provide a comprehensive picture of the performance of various choices of model and ensemble sizes on the Cifar datasets.
Nevertheless, the proposed method is not compared to any existing models and regularization approaches which are applicable to small datasets, including the cited references or other approaches (e.g. [1-4]). This makes the statement of "improving the state-of-the-art" questionable. The evaluation also does not seem to perform adequate cross-validation (e.g. the authors use the "best test performance" across any epoch).

I thus encourage the authors to perform a more comprehensive evaluation of the proposed approach, and further compare to other methods. Some confidence estimates for comparing methods would also be useful, as such small datasets may lead to large variance across different choices of samples. Also, it would be interesting to see how the approach performs beyond just the Cifar dataset, perhaps in other domains where data is more scarce. Other ways to control for computation in each model of the ensemble would be interesting, e.g. how would controlling width instead of depth affect performance?

[1] Arora et al "Harnessing the power of infinitely wide deep nets on small-data tasks."
[2] Bietti et al. "A Kernel Perspective for Regularizing Deep Neural Networks"
[3] Drucker and Lecun "Improving generalization performance using double back-propagation"
[4] Miyato et al. "Virtual Adversarial Training: A Regularization Method for Supervised and Semi-Supervised Learning"

---

> ### Author Response · Authors · 2020-11-25
> **Answer to Reviewer 2**
>
> We would like to thank the Reviewer for the constructive comments and we appreciate that finds our approach promising. We performed a more comprehensive evaluation as suggested:
>
>
> - We empirically show the improvement with respect to the state of the art by comparing our ensembles with [Arora et al. 2020] and [Bietti et al. 2019] following the same evaluation protocol of the original papers (Table 1). Note also that in [Bietti et al. 2019] authors kept a held-out training set for tuning hyper-parameters while, in our case, we didn't do it and left default parameters.
> We definitely share the same objective and experimental evaluation with these two papers. Therefore, we agree on the fact that ours and their approaches are directly comparable.
> Instead, approaches proposed in [Drucker and Lecun]  and [Miyato et al.] were not evaluated and tested on small image datasets falling outside the aim of our experimental evaluation.
> For what concerns evaluation, we used the same evaluation proposed in [Arora et al. 2020] repeating the same experiment multiple times (i.e. different sub-sampled splits for each N) and averaging the test accuracy obtained in each run (i.e. the best testing accuracy scored by the model across the epochs).
>
>
> - We provided in the first submission and in this new version confidence estimates in terms of standard deviations for the different runs in all tables and plots. We added two more datasets (SVHN and Stanford Dogs), we ran several experiments concerning the variation of depth and width (Tables 2 and 5) and we discussed the results in Section 5.3.
>
> [Arora et al. 2020] "Harnessing the power of infinitely wide deep nets on small-data tasks."
>
> [Bietti et al. 2019] "A Kernel Perspective for Regularizing Deep Neural Networks"
>
> [Drucker and Lecun] "Improving generalization performance using double back-propagation"
>
> [Miyato et al.] "Virtual Adversarial Training: A Regularization Method for Supervised and Semi-Supervised Learning"

---

### Official Review · AnonReviewer3 · 2020-10-28
**The paper is missing explanations of the observed phenomena**

**Rating:** 5
**Confidence:** 5

**Review:**

Summary:
In this paper, the authors provide a series of experimens where they show that when dealing with a very small dataset, a single very deep network is outperformed by an ensemble of multiple more shallow networks. More specifically, the authors artificially create training sets from CIFAR10 and CIFAR100 datasets where the number of images per category is limited to 10-250 samples. Then, they compare the test performance of ResNet101, an ensemble of 5 ResNet 20 and an ensemble of 20 ResNet8, trained for classification with different loss functions, i.e. cross-entropy and cosine distance. The bottom line is that the ensembles work better and have a comparable computational complexity in FLOPs.

Strengths:
- The topic of the paper fits well in the paradigm of representation learning.
- The work demonstrates that the community does not have a good understanding of what kind of models must be used when little data is available for training and brings attention to classical techniques for variance reduction.

Weaknesses:
- The paper is basically a compilation of experiments with no explanations of the observed phenomena. The authors perform a set of experiments with already known methods and merey propose the reader to look at the results. I would like to know not only that we need to do ensembles of small networks but also why these ensembles are more efficient than a single deep network in the low data scenario. Why do we observe the difference between using cross-entropy and cosine losses, depending on the network, dataset and its size?
- The novelty of the paper is limited. It is already known from [1] that using ensemble methods in few-shot problems helps the performance a lot. Even if the authors propose a different evaluation strategy, referencing existing work in this field is still required.
- Abblation studies are missing. To be more convinced by the experiments I would like to see how the performance differes if you vary the ensemble size and the single network's capacity. That may improve our understanding of the phenomena too. Experimenting with more datasets may help to answer the question of why the behavior of different loss functions is so different between the two used datasets.
- A question of wheather a single ResNet101 with vanilla training is a fair baseline. It’s been known [2] that to achieve better results on a small-data task it is beneficial to train deeper networks with proper regularization rather than shallow networks. Using an ensemble of N networks is identical to using a single network where each layer is N times wider; each convolutional layer will have N times more filters (that could be obtained by concatenating the weights of the original network), however the convolution operation now changes from a standard to a grouped one (has N groups). The resulting output of the fused network must be averaged across the groups to match the ensemble definition exactly. The group-separated convolutions restrict representational power of the network and introduce stronger regularization, which is most likely the reason for the ensemble to perform better. If we speak about regularizing ResNet101 what kind of regularization did you introduce to adapt it to the small size dataset? It is possible that vanilla training with higher weight decay and more data augmentation is not actually efficient in the case of ResNet101 on the small datasets. Instead, it may require introducing more aggressive data aufmentation [3,4] or some structural changes must be introduced, as for example in [5].


Even though the direction of reserch is interesting and deffinitely useful for the community the work still needs development to be recommended for acceptance.


[1] - Dvornik et.al "Diversity with Cooperation: Ensemble Methods for Few-Shot Classification"
[2] - Geiger at.al "The jamming transition as a paradigm to understand the loss landscape of deep neural networks"
[3] - DeVries et.al "Improved Regularization of Convolutional Neural Networks with Cutout"
[4] - Zhang et.al "Mixup: beyond empirical risk minimization"
[5] - Gastaldi "Shake-Shake regularization"


-----------------------------------------------------------------------------------------------------------------------------------------------------------------------------------
Update after the author's comment:

I appreciate the effort of the authors to add more experiments that all suggest that the ensembles tend to perform better in the small data regime. This makes the case stronger and the story more compelling, hence I raise my score. However, the paper is still missing the core explanations or a hint of why this may be happening, hence I still can not recommend the paper for acceptance.

---

> ### Author Response · Authors · 2020-11-25
> **Answer to Reviewer 3**
>
> We thank the Reviewer for the multiple suggestions and comments. We definitely agree on the fact that the community does not have a clear idea on how to tackle the problem of learning from a small sample since this is a rarely experimented domain, yet very important in practice.
> For this reason, we fairly believe that the results reported in this revised version of our paper can be exploited by the community both as baselines for future works and as a practical guide for solving similar problems.
> We made several additions and changes to address the weakness of our paper:
>
> - In Section 5.4 we motivated the better sample efficiency of ensembles following models proposed in [Geiger et al. 2019] and [Geiger et al. 2020] that we noticed to be in accordance with our experimental findings.
> We also motivated the difference in performance among the two losses and provided a hypothesis following the model proposed in [Geiger et al. 2019].
>
>
> - We added the reference to work [Dvornik et al.] since this is a use of ensembles in a somewhat related domain. However, we kindly point out that few-shot learning and learning from a small sample are implemented by the community in two different ways as we wrote in the introduction. Despite their names are semantically similar, in few-shot learning we still exploit a large base set from which it is possible to learn, “as we know it with big and large nets”, good representations that then we transfer to more tiny and difficult problems. In our scenario, the lack of data from the beginning presents different challenges that are still not clear to the community, as also noted by the Reviewer, and could not be encountered in few-shot learning. Therefore, in our opinion, it is not straightforward to assume that good performance of deep ensembles in few-shot learning would imply the same performance with small datasets.
>
>
> - We added several ablation studies regarding the variation of depth, width, and a number of nets in an ensemble. Refer to Tables 2 and 5 and, in particular, to Section 5.3. Further, we added two more datasets (SVHN and Stanford Dogs) to get more solid empirical evidence.
> It is reasonable to doubt if a ResNet-110 is a fair baseline on small datasets. To better compare our results, we considered several more network architectures and layouts as baselines: ResNet-110, VGG-9, DenseNet-BC-52, and DenseNet-BC-121; we varied the depth of baseline ResNets (26, 50, and 110 layers); we added results of shallower and wider networks (ResNet-8-16, ResNet-8-36, ResNet-8-50, ResNet-8-72; VGG-5-32, VGG-5-76; DenseNet-BC-16 (k=12,30), DenseNet-BC-62 (k=32,56)).  All the above architectures and layouts have been compared with the corresponding ensemble of networks of the same family with the same computational budget.
> In almost all cases, ensembles outperformed the deeper/wider variants, with clear margins, especially over deeper nets (Tables 2, 4, 5).
>  As also suggested by the Reviewer, we made a test with stronger data augmentation on CIFAR-10 adding color distortion and random erasing but still obtained similar trends (Table 5).
> We motivate these consistent results following models proposed in [Geiger et al. 2019] and [Geiger et al. 2020] that we noticed to be in accordance with our experimental findings.
>
> [Dvornik et al.] "Diversity with Cooperation: Ensemble Methods for Few-Shot Classification"
>
> [Geiger et al. 2019] “Jamming Transition as a Paradigm to Understand the Loss Landscape of Deep Neural Networks.”
>
> [Geiger et al. 2020] “Scaling Description of Generalization with Number of Parameters in Deep Learning”.

---

### Official Review · AnonReviewer1 · 2020-10-29
**Official Blind Review #1**

**Rating:** 4
**Confidence:** 4

**Review:**

This paper proposes an ensemble learning method for deep networks in the low data regime. Specifically, the authors empirically compare several ensemble configurations by varying the complexity of base members given a total fixed computational budget. Experiments are conducted on CIFAR-10 and CIFAR-100 datasets. It shows that good results are obtained by keeping low the complexity of single models and increasing the ensemble dimension.

Paper strengths:
+ The paper is well written and organized. It is easy to follow.
+ The topic, i.e., ensemble learning for deep networks, is interesting and deserves further studies.

Paper weaknesses:
- The novelty of this paper is low. No significant technique contribution is performed in this paper. The proposed method is a simple weighted average ensemble. Additionally, the loss functions employed were developed in previous work, which are also not the contributions by the authors. Thus, does it seem an empirical study paper?
- If it is an empirical paper, the experiments are also weak. Only two small-scale (w.r.t. deep learning) datasets, i.e., CIFAR-10 and CIFAR-100 are conducted for evaluation comparisons. Meanwhile, only the ResNet family is utilized as the backbones. It is encouraged to perform comprehensive experiments on large-scale, diverse (e.g., object-centric data and scene-centric data) vision datasets, as well as different network architectures (e.g., ResNets, VGGs, MobileNets, etc). Current experimental results are not sufficient enough to support the conclusions of this paper.
- More analyses are required, such as attempting to reveal why the observations in this paper can happen? In addition, is there a possibility that few training data cannot support the training of big networks (e.g., ResNet-110), rather than the effectiveness of the proposed ensemble process? Thus, small data trained with small networks (e.g., ResNet-8) can achieve better results since small networks can be trained until parameter convergence.

Minor issues:
- The references are not formal. For example, several references have inconsistent formats, e.g., "In Proceedings of the 28th International Conference on machine learning (ICML-11), " of [Deisenroth and Rasmussen, 2011] vs. "In international conference on machine learning," of [Gal and Ghahramani, 2016].
- There are also several typos in this paper. The authors should carefully proofread the paper.

---

> ### Author Response · Authors · 2020-11-25
> **Answer to Reviewer 1**
>
> We’d like to thank the Reviewer for the positive comments regarding the organization, structure, and topic of our work.
> We made several additions and changes to address the weakness of our paper:
>
> - We agree on the fact that our ensemble method is not a novel approach since we have purposely chosen to use a well-known technique. However, in regard to deep ensembles with small data, this paper is the first one to perform a comprehensive analysis and detailed experimental evaluation. More details are given also in the reply to Reviewer 3.
>
>
> - We added more experiments on two more datasets (SVHN and Stanford Dogs) and two more architectures (VGG and DenseNet).
>
>
> - We agree on the fact that ResNet-110, at a first sight, could be considered as a weak baseline in the case of small datasets.
> In reply to this suggestion (also shared by other reviewers), we added tests with other baselines, i.e., networks of different depths and widths. More details are given in the reply to Reviewer 3.
> In almost all tested comparisons between ensembles and deep/wide networks, including variations in network layouts, datasets, and data augmentation, ensembles remain the most performing models (new Tables 2, 4, 5 in the revised submission).
> In Section 5.4 we motivated the better sample efficiency of ensembles following models proposed in [Geiger et al. 2019] and [Geiger et al. 2020] that we noticed to be in accordance with our experimental findings.
>
>
> - Minor issues: we adjusted references and corrected typos.
>
>
> [Geiger et al. 2019] “Jamming Transition as a Paradigm to Understand the Loss Landscape of Deep Neural Networks.”
>
> [Geiger et al. 2020] “Scaling Description of Generalization with Number of Parameters in Deep Learning”.

---

### Official Review · AnonReviewer4 · 2020-11-01
**Through experiments that are consistent with prior literature, but unclear if findings are novel.**

**Rating:** 5
**Confidence:** 5

**Review:**

This paper tackled a studying the effect of ensembling neural networks to particularly improve accuracy in the low data regime.  The paper is well laid out and the experiments are somewhat well motivated as, much experimental study in DL has been around *larger and larger* datasets. The paper seems well written and the experiments are well constructed with means and standard deviations reported for every experiment. These experiments do match prior subsampling experiments literature (at least for Cifar-10).
One minor call out I'd make is that Shankar Et al  (https://arxiv.org/pdf/2003.02237.pdf) shows that a non ensembled CNN has similar performance to a CNTK on subsampled Cifar-10.

My primary problem with this work is a the somewhat expected nature of the findings and the limited scope of the experiments. First of all while Cifar-10 and Cifar-100 are great datasets that researchers should benchmark on, but for this line of work I would have liked to see more. How do these trends look like on ImageNet? What about for non vision tasks, does something similar happen on SQUAD.

Furthermore I'd encourage the authors to explore the tradeoff between number of ensembles vs depth of network more. I'd like to see a 2/3d plot with depth/size of network on one axis, and number of ensmbles on the other, how does the tradeoff frontier look like? It is simply not convincing to just look at a resnet20 vs resnet-8.

I also find the loss comparison direction less interesting than the tradeoff between depth/width/number of ensembles.

Anyway the reason for my score is that this seems like a promising direction but looks like early work. With a couple more datasets and a more thorough experimentation section I'd accept this paper.

---

> ### Author Response · Authors · 2020-11-25
> **Answer to Reviewer 4**
>
> We appreciate that the Reviewer finds our work promising and well-motivated. We thank the reviewer for the provided suggestions that we addressed in the revised version of our manuscript:
>
> - We added two more popular datasets (SVHN and Stanford Dogs). We added more vision tasks since our focus regards image classification and the computer vision field. It would be definitely interesting to try our approach to non-vision tasks. We left it as future work.
>
>
> - We explored more the trade-off between depth and ensembles dimension. More in detail, we varied the depth of the baseline network and the number of ensembles accordingly. We invite the Reviewer to refer to Figure 1 and Table 5 for the results. Moreover, we trained diverse architectures at different depths and compared them with the corresponding ensembles (Table 2).
>
>
> - We also added comparisons for the width dimension. More details are given in the reply to Reviewer 3. In the revised version of the paper, the results of this analysis are in Tables 2, 5, and Figure 1. We discussed these results in Section 5.3.

---

### Author Response · Authors · 2020-11-25
**General comments**

We appreciated all the comments from the reviewers that allowed us to extend the experimental analysis and consequently to better support our claims. In particular, we agree that a more comprehensive evaluation is important to assess the improved performance of ensembles of NN with small datasets. Therefore, we executed many additional experiments, fully reported in the paper and in the appendices. As explained more in detail in the replies to the reviewers below, all the performed experiments confirm that ensembles of NN outperform other methods when considering different datasets, different base NN models, and different layouts of the NN.
Moreover, in comparison with previous literature published on learning with small data sets, the revised version of our paper provides a more comprehensive analysis of the problem addressing many experimental variabilities. We thus believe that this paper will be very relevant for the community addressing this problem in future works.

---

### Decision · Program_Chairs · 2021-01-07
**Final Decision**

**Decision:**

Reject

**Comment:**

The paper received negative and borderline reviews. The reviewers have raised several concerns about the novelty of the approach and the lack of convincing experiments. The rebuttal only partially addresses these concerns. Overall, the area chair agrees with the reviewer's assessment and follows their recommendation.